# RECAST YOUR INPUT VIA A MAPPING FUNCTION FOR ALIGNMENT

## ABSTRACT

Alignment is promoting its critical role among the large language model (LLM) scenarios, which ensures safety, controllability, and trustworthiness of the generation. The popular alignment methods, that is, reinforcement learning from human feedback (RLHF), direct preference optimization (DPO) and such series, usually change weights of the model by elaborate algorithm. Nevertheless, they suffer from the compute drain for training, especially when the parameters' size getting huge. Worse still, people typically do not have access to the weights of the SOTA models, such as GPT-4, which consequently renders the aforementioned algorithms unimplementable. In this paper, we propose to employ a separate LM as the **Refiner**, an input mapping function essentially, to transform the original query into a novel formulation that impels the final generation to align with the expectations. During optimization, an evolution strategy, namely **CMA-ES**, is leveraged to fine-tune the LM with linkage to the generation model. We conduct extensive experiments on various refiner and generation types, and achieving surpassing results.

## 1 INTRODUCTION

Aligning LLM with human preference has consistently proven to be essential for majority of applications, which guarantees it generation authenticity and morality, and circumvents pitfalls of overconfidence Tian et al. (2024); Ethayarajh et al. (2024). Distinguished from the supervised fine-tuning (SFT) process, researchers usually refer to preference data for alignment, with rendering the disparity inside the answer list conspicuous. Typical aligning methods, namely RLHF Christiano et al. (2017) and DPO Rafailov et al. (2023), derive from maximizing the generation return and minimizing the Kullback-Leibler (KL) divergence from original distribution, and vary in implementation types, i.e., reinforcement learning (RL) and contrastive learning (CL). Severing as a critical part for post-training, they shed light on the performance improvement among popular LLMs, like GPT-4 OpenAI et al. (2024) and DeepSeek-R1 DeepSeek-AI et al. (2025). Unfortunately, the

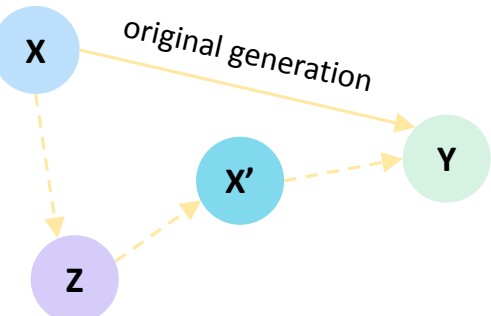

Figure 1: The probability graph of refined generation, i.e., **X->Z->X'->Y**, for alignment where **X->Y** denotes the original generation, **Z** and **X'** represent the latent variable and refined input separately.

aforementioned techniques suffer from huge computational burden when confronting LLMs characterized by a vast parametric ensemble. With exacerbating the predicament, the acquisition of model's weights is usually unattainable for some SOTA LLMs, such as GPT-4 and Claude-3.5 Claude.ai (2024), which results in the infeasibility of these training methodologies.

Black-box prompt optimization (BPO) Cheng et al. (2024) is proposed to steer the input to accommodate the generation LLM, hence evades the training issues discussed above. However, BPO learns its prompt preference optimizer, a relatively small language model (SLM), in isolation from the final generation process, which covers no guarantee that the training datasets they construct

are universally applicable across models. Additionally, in the course of data collection endeavors, BPO commences with the preference data pairs to derive the preference reason and "un-prefer $\mapsto$ prefer" shift fashion, capitalizing on the critical faculties of the LLMs. Notwithstanding it, given hallucination Bouyamourn (2023); Xu et al. (2024) residing within the LLMs, the preference reason cannot entirely supplant the preference pair for training of the preference optimizer.

In light of the preceding deliberations, we propose the input **Refiner** (being analogous to the prompt preference optimizer of BPO), and learn it by interacting with the generation model via *Covariance Matrix Adaptation Evolution Strategy* (**CMA-ES**) Hansen et al. (2003); Sun et al. (2022); Wang et al. (2024). As is displayed in Figure 1, $\mathbf{X} \to \mathbf{Y}$ represents original generation process, from query to the answer. **Note that since we are optimizing a black-box model, gradient descent cannot be directly applied to train** $p(y|x)$. Therefore, we decompose it into a joint stages pair, of which the refinement ($\mathbf{X} \to \mathbf{Z} \to \mathbf{X}'$) models with a latent variable, covering information from the preference pair, and the generation ($\mathbf{X}' \to \mathbf{Y}$) servers as a black-box model that criticizes the refinement. Our contributions are summarized as follows:

• We devise the input refinement module by introducing a latent variable to absorb information from preference pair and reason, which may demonstrates diversity among scenarios.

• We consider the generation part as a black-box model, and utilize CMA-ES method to revise the refinement result for a better adaption to the generation dynamics.

• To enhance the stability within the learning and optimization processes, we introduce a series of adaptive measures, encompassing *posterior regularization* (to leverage preference pair into the refiner part) and *gradient projection* (to ensure quality of the refiner output).

## 2 RELATED WORK

To align LLMs with human intents and preferences, various tuning and infering strategies have been proposed. Prevalent alignment approaches can be summarized into three categories.

**RLHF and DPO.** Existing typical methods of steering LMs to match human preferences include RLHF Christiano et al. (2017), DPO Rafailov et al. (2023), and their variants Meng et al. (2024); Pal et al. (2024). RLHF methods learn a reward model from a curated dataset of human preferences and then use it to optimize a language model policy by RL algorithm, to generate responses assigned high reward, and using KL-penalty to keep the policy from deviating too far from the original model. RLHF has been applied to many prominent language models, and has been shown to improve performance across a wide number of capabilities, including instruction following Ivison et al. (2023) and reasoning Trung et al. (2024). Despite the widespread use and potential of this learning paradigm, aligning LLMs through RLHF remains challenging due to training instability. DPO bypasses the need for explicit reward model and implicitly optimizes the same objective as existing RLHF algorithms (reward maximization with a KL-divergence constraint), which is simple to implement and straightforward to train. However, these post-training methods suffer from huge computational burden and cannot be proceeded further on a closed source LLMs.

**Prompt Optimization.** A different perspective of alignment is to optimize user prompts to suit LLMs' input understanding better, so as to best realize users' intents without updating LLMs' parameters. BPO Cheng et al. (2024) fit a prompt optimizer to a dataset of human preference comparisons and then utilize it to steer human prompts to accommodate LLMs' understanding. In a broad sense, automatic prompt engineering Pryzant et al. (2023); Yang et al. (2024) can also be considered as an input side alignment approach. These methods perform alignment in language space, however, language space may not always be optimal for LLMs' understanding. For example, most word tokens are primarily for textual coherence and not essential for specific even implicit preferences.

**Inference-time Alignment.** Inference-time alignment refers to those procedures that change the decoding strategy to perform alignment directly. One of them is the Best-of-N method. Best-of-N Stiennon et al. (2020); Sessa et al. (2024)) generates N responses for a single prompt, and the best response is selected based on the evaluation of a reward model that measures the suitability of the responses. It is as effective as the state-of-the-art post-training procedures, however, Best-of-N requires vastly more resources at inference time than standard decoding strategies, which makes it computationally not viable. To address this, a computationally-viable inference-time alignment

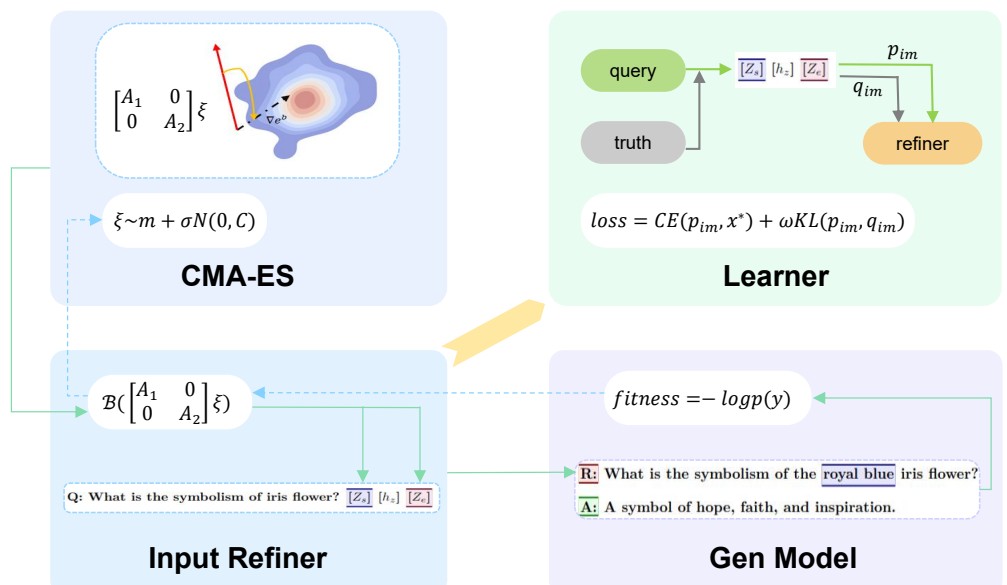

Figure 2: Referring to **Input Refiner** modeled with the latent variable, the initial query is reconfigured into a formulation more aligned with the answer generation process ("bottom-left $\rightarrow$ bottom-right"). We utilize **CMA-ES** to optimize refinement results, severing as pseudo labels for refiner, to accommodate the generation dynamics ("bottom-right $\rightarrow$ bottom-left $\rightarrow$ top-left"). During learning process of input refiner ("top-right"), the posterior regularization is deployed to incorporate information from preference pairs thus enhance the refinement efficacy.

algorithm, Speculative Rejection Sun et al. (2024), is proposed and demonstrated generating high-scoring responses comparable to Best-of-N, while being between 16 to 32 times more computationally efficient.

## 3 PROPOSED METHOD

Referring to the generation process, i.e., $p(y|x)$, we conduct dissociation from a specific language model of "query $\mapsto$ answer" generation and further supply the input refinement with explicitness. Specifically, we decompose original generation via the Bayesian method as the combination of *input refinement* and *answer generation*:

$$p(y|x) = \sum_{(z,x')} p(y|x,z,x')p(z,x'|x) = \sum_{(z,x')} \underbrace{p(y|x')}_{\text{Gen}} \underbrace{p(z|x)p(x'|z)}_{\text{Input Refiner}}, \quad (1)$$

where the input refiner adapts the original input $x$ into $x'$, which will better stimulate the LLM's capacity for for an optimal alignment. $p(y|x')$ servers as the final generation process which generates preferred output after the refined input. The latent variable, i.e., $z$ in the equation, stands for an unobservable effect, namely user preference with diversity Kobalczyk et al. (2024); Qu et al. (2024) and reasoning paths among scenarios, which entrains generation varying potentially.

**Proof** : As is demonstrated in Figure. 1, the joint distribution for variables $(x, z, x', y)$ is expressed as:

$$p(x, z, x', y) = p(x)p(z|x)p(x'|x, z)p(y|x, z, x'). \quad (2)$$

In accordance with their "**head-to-tail**" connection attribute, the distribution will be simplified as:

$$p(x, z, x', y) = p(x)p(z|x)p(x'|z)p(y|x'). \quad (3)$$

Hence, we derive that $p(y|x, z, x') = p(y|x')$.

Similarly, it satisfies that:

$$p(x, z, x') = p(x)p(z|x)p(x'|z). \tag{4}$$

Therefore, the posterior for variables $(z, x')$ is decomposed as: $p(z, x'|x) = p(z|x)p(x'|z)$.

---

**Algorithm 1:** Iterative Optimization

**Input** : $\mathcal{D}, \mathcal{D}_{es}, S, LM_{ir}, LM_{gen}, \mathcal{P}_x, \mathcal{P}_{xy}, \xi, \lambda, i_{es}, A_0, A_1$
**Output** : $LM_{ir}$

1 **DEF** `Init()`:
2     Input refiner process, with only original input: $p_{im}(.) = LM_{ir}(p = \mathcal{P}_x(x))$
3     Posterior regularization, with the added output: $q_{im}(.) = LM_{ir}(p = \mathcal{P}_{xy}(x, y))$
4     The response generation process: $f(.) = LM_{gen}()$
5     Initialization of CMA-ES process: $es$=CMA$(\xi, popsize=\lambda, iter=i_{es})$
6 **DEF** `Emb(ξ)`:
7     Embedding derivation of the input refiner model: $e(.) = LM_{ir}\text{-}Emb()$
8     Update for embeddings of the special tokens $[Z_s]$ and $[Z_e]$:
$$\begin{bmatrix} e_0 \\ e_1 \end{bmatrix} = \begin{bmatrix} e([Z_s]) \\ e([Z_e]) \end{bmatrix} + \mathcal{B}(\begin{bmatrix} A_0 & \mathbf{0} \\ \mathbf{0} & A_1 \end{bmatrix} \xi)$$
9     **return** $e_0, e_1$
10 **DEF** `CMA-ES()`:
11     **while** *not es.stop()* **do**
12        Draw samples of $\xi$ from a normal distribution $N(m, \sigma^2 C)$: $\xi \sim m + \sigma N(0, C)$
13        $e_0, e_1$ =`Emb(ξ)`;   $r = 0$
14        **for** $(x, y) \in D_{es}$ **do**
15           Derive feedback signals from the generation:
            $r \leftarrow r + \mathbf{CE}(f(p_{im}(x, E([Z_s]) = e_0, E([Z_e]) = e_1)), y)$
16        **end for**
17        $r \leftarrow r/|D_{es}|$
18        Update CMA-ES parameters: $\xi, m, C \leftarrow es(\xi, m, C, r)$
19     **end while**
20     **return** $\xi$
21 **while** $s_i \leq S$ **do**
22     Initialize input refiner, posterior regularization, generation and the CMA-ES: `Init()`
23     $\xi^* =$ `CMA-ES()`;   $e_0^*, e_1^*$=`Emb(ξ*)`
24     ### Get pseudo refiner output and train the refiner model:
25     **for** $(x, y) \in D$ **do**
26        $x^* = p_{im}(x, E([Z_s]) = e_0^*, E([Z_e]) = e_1^*)$
27        $l_{rm} = \mathbf{CE}(p_{im}(x), x^*) + \omega \cdot \mathbf{KL}(p_{im}(x), q_{im}(x, y))$
28        $LM_{ir} \leftarrow LM_{ir} - \alpha \nabla l_{rm}$
29     **end for**
30 **end while**

---

### 3.1 INPUT MAPPING FUNCTION

The *input refiner* part in Equation 1 constitutes "$p_{im}$" for abbreviation in this paper. Inspired by Hao et al., we devise $p_{im}$ as a revised autoregressive model, with the latent variable $z$ emerging following the input $x$ as a "soft prompt" Lester et al. (2021). Additionally, a special token pair, i.e., $[Z_s]$ and $[Z_e]$, is introduced to enclose the latent variable for its position marking. As a consequence, we specify the encoder-decoder item, that is, $p_{im}$, in a joint manner.

$$p_{im} \triangleq p(z, x'|x) = p(z|x)p(x'|z),$$
$$\textbf{s.t. } z = H(E(\mathcal{P}_x(x)) \oplus E([Z_s])), \ x' \triangleq \{t_{oi}|i \in [1, L]\} = D(H(z \oplus E([Z_e]))), \tag{5}$$

where operators of $E(.)$ and $H(.)$ are for input text encoding and hidden state calculation in the autoregressive LM, correspondingly. $D(.)$ represents the decoding process. $t_{oi}$ is the token at $i$ position for the refined input $x'$ and $L$ represents the generation length. $\mathcal{P}_x(.)$ is the prompt design

for original input $x$. **The latent variable $z$ is represented as the hidden state of the original input $x$ under the autoregressive LM, which is then fed back into the LM in the form of a continuous prompt to generate the refined input** $x'$. Notably, we lack any ground-truth labels for $x'$, making direct optimization of $p_{im}$ infeasible. To mitigate this unlabeled-data challenge, we adopt a dual strategy: (1) employing posterior regularization to constrain the optimization space (Equation 6); (2) leveraging the CMA-ES algorithm to extract approximate $x'$ values from feedback signals of generation process, i.e., **Gen** in Equation 1, which are then utilized as pseudo-labels $x^*$ (Equation 11).

## 3.2 POSTERIOR REGULARIZATION

Conspicuously, introduction of $z$ in Equation 5 injects supervised signal with scarcity which will conduce to an unstable training of $p_{im}$, especially when $x'$ being not labeled.

With adding more information to estimate $p_{im}$ that satisfies the posterior regularization, we introduce $q_{im}$ to approximate the input refinement process and derive:

$$q_{im} \triangleq p(z, x'|x, y) = p(z|x, y)p(x'|z), \tag{6}$$

where $y$ transfers the output information ahead as an auxiliary for the unobservable effect summarization. For the sake of its modeling, we employ a target network with the fixed parameters from $p_{im}$, and distinguish the output distribution via the input prompt modification. Consequently, within the framework of $q_{im}$, the latent variable $z$ admits the representation of:

$$z = H(E(\mathcal{P}_{xy}(x, y)) \oplus E([Z_s])), \tag{7}$$

where $\mathcal{P}_{xy}(.)$ is a well-designed prompt for incorporation of original input $x$ and the output $y$, which supplies sufficient signals on the reason why original output is optimal. For the preference data, it highlights the rank between preference pair (details are shown in Table 1). During deployment, the training performance of $p_{im}$ is enhanced through minimization of distributional discrepancies (e.g., KL divergence) between $p_{im}$ and $q_{im}$.

## 3.3 OPTIMIZATION OBJECTIVE

With freezing parameters of the **Gen** in Equation 1, we regard it as a black-box model, denoted by $f(.)$, for the whole framework construction, and formulate the optimization as:

$$min \ \mathcal{L}(f(p_{im}(x)), y). \tag{8}$$

Given the inaccessibility of $\nabla f$ from the black-box model, we utilize an evolution strategy, namely CMA-ES, for optimization. Generally, via CMA-ES, a variable updates its value by sampling from a *Gaussian Distribution*, that is $N(m, \sigma^2 C)$. With further considering the truth that CMA-ES usually deals with the variable of limited dimension, we introduce the *Matrix Factorization* on the embedding bias of the latent marking tokens, i.e., $[Z_s]$ and $[Z_e]$ (referring to Equation 5), for updating from the feedback of $f(.)$ and propose an iterative optimization strategy. Specifically, we decompose the embedding $e$ as:

$$\begin{bmatrix} e_0 \\ \overline{e_1} \end{bmatrix} = \begin{bmatrix} e_0^b \\ \overline{e_1^b} \end{bmatrix} + \mathcal{B}(\begin{bmatrix} A_0 & \mathbf{0} \\ \mathbf{0} & A_1 \end{bmatrix} \begin{bmatrix} \xi_0 \\ \xi_1 \end{bmatrix}),$$
$$\textbf{s.t. } e_0^b = E_b([Z_s]), \ e_1^b = E_b([Z_e]) \tag{9}$$

where $E_b(.) \in \mathbb{R}^d$ ($d$ is the embedding dimension) is the **initial embedding function** for the refinement model. $e_i$ functions with adding the *embedding bias* for a stable optimization. $A_i \in \mathbb{R}^{d \times d_z}$ represents the projection matrix and $\xi = [\xi_0, \xi_1]^T$ ($\xi_i \in \mathbb{R}^{d_z}$, $i \in \{0, 1\}$) is the variable updated by CMA-ES. $d_z \ll d$ means the evolution dimension. $\mathcal{B}(.)$ is a constraint function that restricts the embedding bias value to a manageable scope (details are displayed in Equation 13).

For optimizing $\xi_i$, we conduct sampling at the **evolution step** $t$ as:

$$(\xi_i)_j^{(t)} \sim m_i^{(t-1)} + \sigma_i^{(t-1)} N(0, C_i^{(t-1)}), \tag{10}$$

where $j \in [1, \lambda]$ ($\lambda$ means the population size for the evolution strategy) denotes the population index. $m_i^{(t-1)}$ and $\sigma_i^{(t-1)}$ are the expected value and standard deviation over the population at step $t-1$, correspondingly. $C_i^{(t-1)}$ represents the covariance matrix.

At each iteration, we decompose the optimization process into dual stages of which one for the *Pseudo Label Derivation* and the other for the *Refinement Model Fine-tuning* (details are displayed in Algorithm 1).

**Pseudo Label Derivation:**    Due to the absence of annotated $x'$ (Equation 5), we employ CMA-ES to approximate $x'$ through feedback of the generation process. At the beginning of each iteration, we forward the input refiner model, that is $p_{im}$ (seeing at Equation 5), to derive the initial generation tokens, with $\xi$ being the dependent variable for optimization. Furthermore, CMA-ES method is implemented to obtain the optimal refined input, i.e., $x^*$, referring to the black-box generation model $f(.)$, which is regarded as the *Pseudo Label* for $p_{im}$ during the implementation. The fitness expression for CMA-ES at the current generation

| Type | Prompt |
|------|--------|
| $s$ | You are an expert prompt engineer. Help me improve this query to get a more helpful and harmless response. |
| $\mathcal{P}_x$ | $s$ + Please output the modified query only! Query:{ } |
| $\mathcal{P}_{xy}$ | $s$ + Form the judgment upon the following truth. Query:{ } Truth:{ } |

Table 1: Prompt design for both $p_{im}$ and $q_{im}$.

step, i.e., $\mathcal{L}(.)$ in Equation 8, is to calculate the cross-entropy value between $f(p_{im}(.))$ and the ground-truth output $y$. Additionally, the optimization is conducted on a subset of the training data ($\mathcal{D}_{es} \subset \mathcal{D}$).

$$x^* = p_{im}(x, E([Z_s]) = e_0^*, E([Z_e]) = e_1^*),$$

$$\textbf{s.t. } \begin{bmatrix} e_0^* \\ e_1^* \end{bmatrix} = \begin{bmatrix} e_0^b \\ e_1^b \end{bmatrix} + \mathcal{B}(\begin{bmatrix} A_0 & \mathbf{0} \\ \mathbf{0} & A_1 \end{bmatrix} \xi^*), \ \xi^* = argmin_\xi \sum_{(x,y) \in \mathcal{D}_{es}} (\textbf{CE}(f(p_{im}(x, \xi)), y)),$$

$$(11)$$

where $e_i^*(i \in \{0, 1\})$ and $\xi^*$ are for the optimal values, respectively. $argmin_\xi$ is optimized by the CMA-ES algorithm.

**Refinement Model Fine-tuning:**    We fine-tune the refinement model from two aspects: minimizing the cross-entropy between the pseudo label $x^*$ and its corresponding prediction, and invoking the KL-divergence between distributions of $p_{im}$ and $q_{im}$ to regularize them (seeing at Equation 6).

$$l_{rm} = \textbf{CE}(p_{im}(x), x^*) + \omega \cdot \textbf{KL}(p_{im}|q_{im}),$$

$$(12)$$

where $\omega$ is the trade-off factor.

## 4 EXPERIMENTS

In this paper, we conduct discussions on several alignment scenarios with preference datasets, building upon some popular LMs of open access for final generation. In case of the intricate functioning of the black-box model, the vLLM Kwon et al. (2023) architecture is utilized to wrap the generation model for prohibition of parameter accessibility. Additionally, we utilize prompt engineering which converts original input into a refined one (prompt details are displayed in Table 1, where prompt for $q_{im}$ fuses that of $p_{im}$ and information from the preference pair).

**Training details:**    Referring to BPO method, we construct the training datasets from four resources, namely: the **OASST1** dataset Köpf et al. (2023) which possesses response ranks from human-annotated; the **HH-RLHF** dataset Bai et al. (2022) which covers helpfulness and harmfulness responses for human preference; the **Chatbot Arena Conversations** dataset Zheng et al. (2023) collected from the online Chatbot platform; the subset of **Alpaca-GPT4** dataset Peng et al. (2023) with GPT-4 generated preference.

During the experiment, we randomly sample 256 instances from the dataset, which is treated as the optimization set, i.e, $D_{es}$ in Algorithm 1, for CMA-ES method. As for the constraint function, i.e., $\mathcal{B}$

| Gen Model | Pair | | BPO-test | | | Dolly | | | Vicuna | | | Self-instruct | | |
|---|---|---|---|---|---|---|---|---|---|---|---|---|---|---|
| | | | win | loss | tie | win | loss | tie | win | loss | tie | win | loss | tie |
| Llama-8B | ours | ORG | **57.0** | 41.0 | 2.0 | **54.0** | 43.5 | 2.5 | **56.3** | 42.5 | 1.2 | **51.6** | 45.6 | 2.8 |
| | ours† | ORG | **51.8** | 46.0 | 2.2 | **51.4** | 46.4 | 2.2 | **51.3** | 48.1 | 0.6 | **53.3** | 44.7 | 2.0 |
| | ours | BoN | **52.0** | 46.0 | 2.0 | **52.0** | 46.0 | 2.0 | **57.5** | 42.5 | 0.0 | 48.9 | **50.3** | 0.8 |
| | ours† | BoN | 46.5 | **51.0** | 2.5 | **49.5** | 48.5 | 2.0 | **55.0** | 45.0 | 0.0 | **50.8** | 47.2 | 2.0 |
| | ours | SR | **70.5** | 27.5 | 2.0 | **64.0** | 34.0 | 2.0 | **67.5** | 31.2 | 1.3 | **72.6** | 25.8 | 1.6 |
| | ours† | SR | **73.5** | 24.5 | 2.0 | **63.0** | 36.0 | 0.1 | **68.7** | 31.3 | 0.0 | **67.9** | 31.3 | 0.8 |
| | ours | DPO | **49.5** | 47.5 | 3.0 | **50.0** | 47.5 | 2.5 | **55.0** | 45.0 | 0.0 | **51.9** | 47.6 | 0.5 |
| | ours† | DPO | **51.5** | 46.0 | 2.5 | 49.0 | 49.0 | 2.0 | **51.3** | 48.7 | 0.0 | **52.4** | 47.2 | 0.4 |
| | ours | BPO | **50.5** | 44.5 | 5.0 | **56.5** | 41.0 | 2.5 | **62.5** | 37.5 | 0.0 | **50.4** | 47.2 | 2.4 |
| | ours† | BPO† | **60.0** | 36.8 | 3.2 | **55.9** | 40.5 | 3.6 | **58.8** | 38.8 | 2.4 | **59.9** | 38.3 | 1.8 |
| Mistral-7B | ours | ORG | **56.5** | 41.0 | 2.5 | **57.5** | 42.0 | 0.5 | **63.8** | 36.2 | 0.0 | **56.0** | 43.2 | 0.8 |
| | ours† | ORG | **59.5** | 37.5 | 3.0 | **60.0** | 39.5 | 0.5 | **60.0** | 38.1 | 1.9 | **51.8** | 47.4 | 0.8 |
| | ours | BoN | **54.0** | 43.0 | 3.0 | 49.5 | **50.0** | 0.5 | **51.3** | 48.7 | 0.0 | 46.8 | **51.9** | 1.3 |
| | ours† | BoN | **54.5** | 43.5 | 2.0 | **54.5** | 44.0 | 1.5 | 43.7 | **53.7** | 2.6 | 47.2 | **51.2** | 1.6 |
| | ours | SR | **71.5** | 27.5 | 1.0 | **67.5** | 31.0 | 1.5 | **70.0** | 27.5 | 2.5 | **67.8** | 31.3 | 0.9 |
| | ours† | SR | **74.0** | 24.0 | 2.0 | **67.5** | 32.0 | 0.5 | **66.3** | 31.3 | 2.4 | **64.3** | 34.9 | 0.8 |
| | ours | BPO | **58.0** | 39.5 | 2.5 | **60.0** | 39.0 | 1.0 | **57.5** | 42.5 | 0.0 | **52.4** | 45.6 | 2.0 |
| | ours† | BPO† | **61.5** | 38.0 | 0.5 | **58.0** | 39.5 | 2.5 | **62.5** | 35.0 | 2.5 | **59.1** | 38.9 | 2.0 |
| Qwen-14B | ours | ORG | **57.0** | 39.5 | 3.5 | **56.0** | 43.0 | 1.0 | **65.0** | 33.8 | 1.2 | **53.7** | 43.0 | 3.3 |
| | ours† | ORG | **61.7** | 34.6 | 3.7 | **59.0** | 40.1 | 0.9 | **65.6** | 32.5 | 1.9 | **54.8** | 42.2 | 3.0 |
| | ours | BoN | **49.5** | 47.5 | 3.0 | **68.0** | 31.0 | 1.0 | **52.5** | 45.0 | 2.5 | **51.5** | 47.6 | 0.9 |
| | ours† | BoN | **54.0** | 43.5 | 2.5 | 47.5 | **52.0** | 0.5 | **55.0** | 43.8 | 1.2 | 45.2 | **53.2** | 1.6 |
| | ours | DPO | **68.0** | 28.0 | 4.0 | **67.0** | 30.5 | 2.5 | **77.5** | 22.5 | 0.0 | **64.7** | 31.7 | 3.6 |
| | ours† | DPO | **67.0** | 29.5 | 3.5 | **65.0** | 34.5 | 0.5 | **80.0** | 17.5 | 2.5 | **68.6** | 31.0 | 0.4 |
| | ours | BPO | **59.5** | 34.0 | 6.5 | **59.0** | 37.5 | 3.5 | **57.5** | 41.3 | 1.2 | **56.0** | 41.7 | 2.3 |
| | ours† | BPO† | **60.0** | 37.0 | 3.0 | **62.0** | 37.5 | 0.5 | **75.0** | 23.8 | 1.2 | **63.1** | 35.3 | 1.6 |

Table 2: The comparison score(%) for alignment evaluation. Llama-3.2-3b-instruct is employed as the input refiner model (except for †). † means Llama-3.2-1b-instruct is utilized for refinement.

in Equation 9, we devise it in combination with the *gradient descent mechanism*, which ensures a reasonable searching space without bringing about a chaotic generation result:

$$\mathcal{B}(e_{bi}) = -\alpha * \sigma\left(\frac{e_{bi} \cdot \nabla e_i^b}{\nabla e_i^b \cdot \nabla e_i^b}\right)\nabla e_i^b, \tag{13}$$

where $e_{bi} = A_i \xi_i$ is the embedding bias for $e_i^b$ ($A_i$ is the projection matrix in Equation 9), $\sigma(.)$ represents the *sigmoid* function. The item within $\sigma(.)$ is to project the bias value into the gradient direction which maintains a controllable generation initialization for CMA-ES optimization (the schematic representation is shown in the top-left part of Figure 2). $\alpha$ is a coefficient acting in a manner comparable to the learning rate. We calculate $\nabla e_i^b$ by feeding back the loss value, i.e., $\mathbf{CE}(p_{im}, y)$, with freezing other layers of the model. Moreover, by employing the *Total Differential Formula*, it is derived that:

$$\delta(\sigma(g)) = \frac{e^{-g}}{(1+e^{-g})^2} * \frac{\delta(\xi_i) \cdot (\nabla e_i^b \times A_i)}{\nabla e_i^b \cdot \nabla e_i^b}, \;\; \textbf{s.t.} \;\; g = \frac{e_{bi} \cdot \nabla e_i^b}{\nabla e_i^b \cdot \nabla e_i^b}, \tag{14}$$

where the operator $\times$ means the common *matmul product*. Therefore, we initialize the step value, i.e, $\delta_0 = \delta(\xi_i)$, with $\delta(\sigma(g))$ as 0.001. Referring to the population size, i.e., $\lambda$ in Algorithm 1, we set its value in accordance with Hansen & Kern, i.e., $\lambda = 4 + \ln Z_d$, where $Z_d = 2 * d_z$ is the dimension for optimization variable.

For the sake of the truth completion ($q_{im}$ in Table 1), we implement a simple measure which demonstrates the ranks between the preference pair explicitly. The template we employ is:

> The response of "**s1**" is better than that of "**s2**" to resolve the query. $[Z_s]z[Z_e]$.

where the suffix pattern, referring to Equation 5, is to summarize the truth expression in a latent manner.

**Evaluation datasets:** The **BPO-test** dataset is sub-sampled from the constructed data of BPO baseline. **Dolly Eval** is a subset of the Dolly dataset Conover et al. (2023). **Vicuna Eval** is collected

by the Vicuna Team Chiang et al. (2023) amongst eight categories for LLM quality evaluation. **Self-instruct** is introduced by Wang et al., under several manually-written novel tasks for instruction-following finetuning.

**Baselines for comparison:** **BoN** (Best-of-N sampling Stiennon et al. (2020); Sessa et al. (2024)) which selects the highest-score generation amongst $N$ (= 120 for the experiment) candidates according to the reward model; **SR** (Speculative Rejection Sun et al. (2024)) that dynamically decreases the candidates number for sampling efficiency. **DPO** Rafailov et al. (2023) transfers the RL process into a contrastive learning manner. **BPO** Cheng et al. (2024) modifies the input by employing a Seq2Seq model for better alignment. **ORG** demonstrates the comparison results with a direct generation by the original language model.

## 4.1 MAIN RESULTS

We conduct the **win-rate** assessment on three generation models, that is, **Llama-8B** (Meta-Llama-3-8B-Instruct Grattafiori et al. (2024)), **Mistral-7B** (Mistral-7B-Instruct-v0.3 Nadhavjhala & Tong (2024)) and **Qwen-14B** (Qwen2.5-14B-Instruct Yang & et al. (2025)), with the generation quality being evaluated by **Qwen-72B**. Specifically, we supply the evaluator with three distinct options for comparative assessment: A) the former is superior to the latter, B) the latter surpasses the former, and C) both are on par, rendering them indistinguishable. These options are designed to facilitate a nuanced and precise evaluation of a pair of generated outcomes, ensuring a rigorous and objective comparison. As for the refiner, we engineer it with a couple of models of relatively small size, i.e., Llama-3.2-1b-instruct and Llama-3.2-3b-instruct.

As is displayed in Table 2, with devised refiner framework, the final generation demonstrates superiority over that of the correspondingly original model, which varies among model architecture and size. Concretely speaking, the transition from original model to the proposed method, refiner size being 3b, yields an average improvement of 11.58% in win-rate ($win - loss$) across four distinct datasets, with Llama-8B serving as the generation cornerstone. The observed performance enhancements for Mistral-7B and Qwen-14B are at 17.85% and 18.18%, respectively. Our method still dominates for the 1b refiner that the concomitantly results are 5.65%/ 17.2%/ 22.93%.

| Gen Model | BPO-test | | | Dolly | | | Vicuna | | | Self-instruct | | | ↗ |
|---|---|---|---|---|---|---|---|---|---|---|---|---|---|
| | **win** | loss | tie | **win** | loss | tie | **win** | loss | tie | **win** | loss | tie | |
| **Llama-8B** | **53.5** | 45.0 | 1.5 | 48.0 | **49.6** | 2.4 | **51.5** | 45.5 | 3.0 | **51.2** | 48.8 | 0.0 | **3.83** |
| **Mistral-7B** | **53.0** | 44.5 | 2.5 | 48.6 | **49.6** | 1.8 | **49.0** | 49.0 | 2.0 | **51.9** | 48.1 | 0.0 | **2.83** |
| **Qwen-14B** | **51.5** | 47.5 | 1.0 | **52.7** | 45.7 | 1.6 | **55.0** | 42.5 | 2.5 | **53.0** | 45.0 | 2.0 | **7.87** |

Table 3: The comparison score(%) for "W/.(CMA-ES) - W/O.(CMA-ES)" pair. Llama-3.2-3b-instruct is the input refiner.

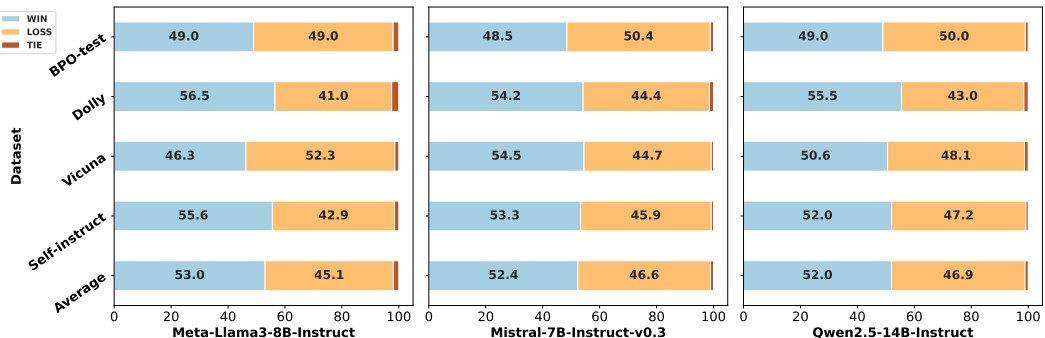

Figure 3: Evaluation on the "W/.($q_{im}$) - W/O.($q_{im}$)" pair, where the refiner is constructed by Llama-3.2-3b-instruct.

In comparison with BPO, our model also showcases superior performance and enhanced capabilities. Delving into the specifics, it attains 12.43% augmentation in the win-rate metric, leveraging 3b size for refiner and Llama-8B for generation. Referring to refiner of 1b, our model spearheads an advancement to score of 20.05%, elucidating that its preeminence over BPO is markedly amplified when deployed with a small size refiner. An exact phenomenon is observed across the other two models as well (15.33% → 22.43% for Mistral with refiner from 1b to 3b, 19.38% → 31.63% for Qwen). This substantiates the premise that **a further optimization of the refiner via interaction with**

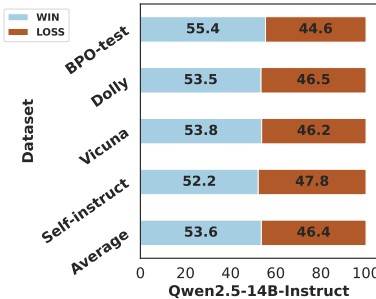

Figure 4: Evaluation on the "Ours-BPO" pair with GPT-4.

**the generation process can indeed enhance its adaptability to the generation dynamics**. It is manifest for refiner of deficient capability. We also employ *gpt-4-turbo-128k* as the evaluator, to judge the results between the propose method and BPO, with Qwen-14B as the answer generator. Referring to Figure 4, our method surpasses BPO by average of 7.2% under the GPT-4 evaluation, with Qwen-14B severing for generation, which also substantiates the preeminence of our model.

When contrasted with SR ad DPO, our method consistently evinces a pronounced superiority, where the improved values are 38.47% and 22.79% averagely among generation architecture and refiner size. It demonstrates comparable efficacy to BoN, e.g., +2.53% for Llama-8B with 1b refiner, and +1.88% for Mistral-7B, nevertheless, BoN necessitates multiple generations at the inference time, each of which is subsequently evaluated and scored to ascertain the optimal output by a reward model. Consequently, this process incurs a substantial expenditure of computational resources, rendering the inference markedly resource-intensive.

It is evident that the training and test data follow different distributions. Methods like DPO require fine-tuning the parameters of the generation LLM, whereas both BPO and our approach only train the input refinement model. Consequently, our methods preserve the original LLM's capabilities and can intuitively achieve stronger performance on out-of-distribution data. This explains the superior results of our method compared to DPO in Table 2.

## 4.2 ABLATION STUDY

**Evaluation on CMA-ES optimization:** We fine-tune the basic model with the latent variable and posterior regularization, and eliminate the CMA-ES module for comparison. Table 3 elucidate impact of this part that being comparison with the complete model, the evaluator prefer to rank worse for this setting. Specifically, the win-rate descents by average 3.83% without the module, with Llama-8B in the generation process. The corresponding results are 2.83% and 7.87% for Mistral-7B and Qwen-14B, separately.

**Evaluation on Posterior Regularization:** For this setting, we ignore the information from the preference pair, i.e., $q_{im}$, and assign 0 to $\omega$ in Equation 12. As is demonstrated in Figure 3, the complete model exhibits a superior win-rate, surpassing that of $q_{im}$ free model by 7.9% for Llama-8B generator. The metrics are 5.8% and 5.1% for other two models. This empirical evidence unequivocally validates the instrumental role of $q_{im}$ in augmenting the performance of the refinement.

## 5 CONCLUSIONS

In the realm of LLM alignment, we innovatively harness Bayesian method to introduce the **input refiner** which functions for query refinement and adaptation to the answer generator. Our refiner model is architected upon a latent variable, meticulously encapsulating the heterogeneity inherent of "input - refinement" pairs and assimilating insights from preference pairs. Moreover, we integrate the **CMA-ES** method to establish a connection between refiner and the generation process, ensuring that the refinement exhibit a heightened congruence with expectations. We conduct experiments on generation models of distinct architecture and size, evincing efficacy of the proposed method.

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

## A  ESSENTIAL NOTATIONS

| Variable | Description | Type |
|---|---|---|
| $x$ | original input | text |
| $z$ | latent variable | vector |
| $[Z_s], [Z_e]$ | position markers for $z$ | token |
| $e_0^b, e_1^b$ | initial embeddings for $[Z_s], [Z_e]$ | vector |
| $x'$ | refined input, derived from the latent variable $z$. | text |
| $x^*$ | pseudo label for $x'$, derived from CMA-ES. | text |
| $\xi$ | the optimizable variable for CMA-ES | vector |
| $y$ | final output, preferable response. | text |

Table 4: Essential notations of the proposed method.

## B  TRAINING SETTINGS

All experiments are designed and executed utilizing NVIDIA A800-SXM4-80GB GPUs, with comprehensive training specifications delineated in Table 5.

| | Description | Refiner | CMA-ES |
|---|---|---|---|
| **N** | training data | 1e4 | 256 |
| $\mathbf{n}_g$ | maximum generation response length | 64 | 128 |
| $\mathbf{d}_z$ | dimension of CMA-ES latent variable, i.e., $z$ | _ | 16 |
| **lr** | learning rate | 1e-5 | 2e-6 |
| **S** | training epochs or optimization steps | 3 | 30 |
| $\mathbf{c}_{kl}$ | KL loss coefficient | 0.02 | _ |
| **bs** | training batchsize | 8 | _ |

Table 5: Training Details for our experimental setting.

## C  EVALUATION PROMPT

---

**1: Evaluation-Aware Prompt**

Please act as an impartial judge and evaluate the quality of the responses provided by two AI assistants to the user question displayed below.

- - - - - - - - - - - - - - - - - - - - - - - - - - - - - - - - - - - - - - - - - - - - -

You should choose the assistant that follows the user's instructions and answers the user's question better. Your evaluation should consider factors such as the helpfulness, relevance, accuracy, depth, creativity, and level of detail of their responses.
Begin your evaluation by comparing the two responses and provide a short explanation. Avoid any position biases and ensure that the order in which the responses were presented does not influence your decision. Do not allow the length of the responses to influence your evaluation. Do not favor certain names of the assistants. Be as objective as possible.

- - - - - - - - - - - - - - - - - - - - - - - - - - - - - - - - - - - - - - - - - - - - -

After providing your explanation, output your final verdict by strictly following this format: ̈[[A]] ̈if assistant A is better, ̈[[B]] ̈if assistant B is better, and ̈[[C]] ̈for a tie."

---

"prompt template":
"[User Question]{question}[The Start of Assistant A's Answer]{answer a}[The End of Assistant A's Answer][The Start of Assistant B's Answer]{answer b}[The End of Assistant B's Answer]",
"description": "Prompt for general questions",
"category": "general",
"output format": "[[A]]"

## D  LIMITATIONS

In this paper, we propose a joint strategy that establishes a connection between **input refiner** and the answer generation process via **CMA-ES** algorithm. With it, initial refinement results will get optimized and adpated to the generation dynamics, hence derive a better aligned expression.

The method still harbors minor imperfections that necessitate improvement. For instance, the CMA-ES algorithm exhibits a dependency on initial values (despite the incorporation of certain robustness measures within the algorithm, i.e., the gradient project and initial step size in Equations 13 and 14). Additionally, the data sampled ($\mathcal{D}_{es}$) for the CMA-ES process can also influence the efficacy of the optimization. These issues warrant further exploration and investigation.

## E  LLM USAGE CLARIFICATION

The application of LLMs in this paper is limited exclusively to polishing the text, particularly in refining specific vocabulary and phrases.

