# OpenReview forum: "Recast Your Input via a Mapping Function for Alignment"
_ICLR.cc/2026/Conference — Submitted to ICLR 2026_

### Official Review · Reviewer_wsCV · 2025-10-31

**Soundness:** 3
**Presentation:** 3
**Contribution:** 2
**Rating:** 4
**Confidence:** 4

**Summary:**

This paper proposes using an additional Language Model (LM) as a Refiner to improve the input, which ultimately leads to outputs with better alignment. Experimental results have demonstrated the effectiveness of this method.

**Strengths:**

1. The method reduces parameters requires training, which is more efficient for preference alignment.
2. The effectiveness of the proposed method is validated through experiments.

**Weaknesses:**

1. The approach bears a strong resemblance to existing methods that improve the performance by refining the prompt, which raises concerns about the novelty of the paper.
2. Ablation studies suggest that the impact of the CAM-ES module is marginal or not statistically significant.

**Questions:**

While the parameter efficiency is a stated advantage, the paper lacks a theoretical justification for a key finding: why does this method outperform full fine-tuning approaches like DPO and BPO? Especially since your experiments seem to confirm this counter-intuitive result, I'm confused about this experimental result.

---

> ### Author Response · Authors · 2025-11-20
> **Discussions on the reviews**
>
> We extend our sincere gratitude to the reviewer for the valuable insights.
>
> ## Weakness 1:
>
> Although our method also refines prompts to enhance alignment, its principal contribution lies in establishing a cohesive link between the traditionally separate processes of refinement and generation. By integrating them into a holistic framework for joint optimization, we significantly elevate the quality of the refinement itself—a critical aspect overlooked by numerous existing approaches.
>
> ## Question:
>
> BPO: While BPO is also a lightweight method that refines the input via a small model to enhance final output quality, its refinement and generation stages remain inherently disjointed. We posit that this decoupling can lead to refined prompts that are suboptimally aligned with the target LLM, thereby explaining its inferior performance compared to our integrated approach.
>
> DPO: A distributional shift exists between the training and test datasets. Both our method and BPO solely refine the input while keeping the generative LLM entirely unchanged. Consequently, although the data distributions differ, the preserved capabilities of the LLM provide a guaranteed level of performance on the test set. In contrast, since DPO is fine-tuned on a dataset that differs significantly from the test distribution, its performance is not guaranteed to surpass that of prompt-based methods.

---

### Official Review · Reviewer_xkVV · 2025-11-01

**Soundness:** 2
**Presentation:** 2
**Contribution:** 2
**Rating:** 4
**Confidence:** 3

**Summary:**

The paper under review proposes a novel approach to align large language models (LLMs) with human preferences by introducing an input refiner module. This module employs a latent variable to transform the original input into a refined version that better aligns with the desired output. The method utilizes the Covariance Matrix Adaptation Evolution Strategy (CMA-ES) to optimize the refinement process, ensuring that the generated responses meet expectations without requiring access to the model's internal parameters.

**Strengths:**

The introduction of an input refiner using latent variables and CMA-ES is a novel contribution that addresses the limitations of existing alignment methods, particularly those requiring access to model weights. By avoiding direct manipulation of model parameters, the proposed method reduces computational overhead, making it feasible for use with state-of-the-art models like GPT-4. The method is adaptable to different LLMs and can be integrated with various generation models.

**Weaknesses:**

1. The concept of alignment through refining prompts is not new and somewhat outdated, as it has been extensively explored in previous literature. The authors should provide a more comprehensive discussion on how this paper distinguishes itself from prior works [1].

2. Though the LLM can only be a black-box module for some closed-source LLMs like GPT-4, as you can revise its input prompt, you can also revise its output response for alignment. Doing them together could be more effective than only doing it in the input side [2].

3. Posterior regularization can conflict with alignment goals if the output y is not aligned with the input x. This misalignment can introduce bias in optimizing the learning of z in Eq. (8). The authors might consider using a preference model to decide whether to apply regularization with y. The current results are not convincing and should be evaluated against more complex alignment benchmarks, such as diverse preference sets.

4. The authors should provide a more detailed explanation in Section 3.1, particularly regarding Eq. (5), which lacks rigor. Variables z and x′ should be sampled from distributions rather than being deterministically projected.


[1] A systematic survey of prompt engineering in large language models: Techniques and applications.

[2] EFFICIENT LLM ALIGNMENT VIA HIERARCHICAL COARSE-TO-FINE REFINEMENT.

**Questions:**

See the weakness.

---

> ### Author Response · Authors · 2025-11-20
> **Discussions on the reviews**
>
> We extend our sincere gratitude to the reviewer for the valuable insights.
>
> ## Weakness 1 & 2:
>
> Thank you for sharing these two works. Our approach indeed shares conceptual common ground with the second paper (coarse-to-fine refinement). Conventionally, most input refinement methods decouple the refinement process from LLM generation, treating them as independent steps. This separation offers no guarantee that the refined input is optimal for the subsequent generation. In contrast, our algorithm bridges these two processes by leveraging the CMA-ES optimizer to feed generation results back into the input refinement module, thereby aligning the refinement more closely with the generative output.
>
> Thus, although our method does not explicitly refine the output, it effectively propagates optimization signals from the output back to the input refinement via CMA-ES, leading to a substantively enhanced refinement process.
>
> ## Weakness 3:
>
> Thank you for your valuable suggestions. We acknowledge that the misalignment between X and Y that you mentioned is not unique to our method but is a challenge inherent to any SFT-based approach. This stems from the fundamental reliance on learning from training data, which inevitably contains noise. Nonetheless, the overarching objective remains to leverage the X-Y alignment present in the data to develop a robust model with strong generalization capabilities. Therefore, achieving this X-Y alignment serves as a foundational premise in our algorithmic design. We fully recognize that the method you proposed is theoretically more sound and reliable. We will actively explore its integration in our subsequent work to drive further optimization.
>
> ## Weakness 4:
>
> The implementation of this component can be described as follows: The input x is first fed into the language model to obtain its hidden state, denoted as z. This hidden state z, which shares the same dimensionality as the model's embedding space, is then treated as a continuous embedding and reprojected into the language model. The model subsequently decodes this representation to reconstruct x'. In this process, the special tokens [Zs] and [Ze] are used to demarcate the positional boundaries of z within the sequence.

---

### Official Review · Reviewer_jAdW · 2025-11-01

**Soundness:** 2
**Presentation:** 1
**Contribution:** 2
**Rating:** 2
**Confidence:** 3

**Summary:**

This paper proposes a method to align black-box large language models (LLMs) by learning a "mapping function" to "recast" the user input. This function is optimized using the CMA-ES algorithm to steer the model towards more aligned responses.

While the paper addresses an important and timely problem, its execution suffers from significant flaws in clarity, methodological explanation, and experimental validation, making it difficult to assess the true merit of the proposed approach.

**Strengths:**

+ The paper tackles a highly relevant and challenging problem: the alignment of proprietary, black-box LLMs where access to model weights is not available.
+ The basic idea of refining the prompt using a small LLM, like 1b/3b, seems to be practical and valuable.

**Weaknesses:**

+ The paper is quite difficult to follow. The writing is disorganized, and key concepts are not introduced clearly. Notations are not defined in the proper place. The methodology section, along with Figure 2, is nearly incomprehensible.
+ The core method is not adequately explained. Algorithm 1 is presented as the main framework but lacks a clear, step-by-step textual explanation to accompany it. The "mapping function" itself is not clearly parameterized, making it hard to understand what is being optimized. The paper would greatly benefit from a simple, concrete "with/without" example to help the reader build intuition for what the input refiner is doing.
+ The figures and tables are not up to publication standards.
  - Figure 2 is low-resolution and difficult to read.
  - Tables are populated with a large number of raw scores but lack clear summary statistics (e.g., averages, confidence intervals). This makes it very difficult to interpret the results or draw conclusions.
+ The experimental setup and analysis are insufficient.
  - The proposed method fails to outperform the `Best-of-N` baseline, a key result that is not discussed.
  - More troublingly, the tables show the *original* base model performing better than both BPO and DPO. This is a highly unusual result that suggests a fundamental problem with the baseline implementations, yet it is presented without comment.
  - The very low "tie" ratio in the pairwise comparisons is also not analyzed, as it does not always appear in alignment papers.
  - The results are evaluated using "gpt-4o-turbo" as an LLM-as-a-judge, which may not be strong enough for the evaluation.
  - With the counterintuitive results, some experimental setups are missing. For instance, the paper uses DPO as a baseline but provides no information on how it was trained.

**Questions:**

How do the authors choose the model for experiments? In the paper, llama 3, llama 3.2, mistral v0.3, and qwen 2.5 with different model sizes are included.

---

> ### Author Response · Authors · 2025-11-20
> **Discussions on the reviews**
>
> We extend our sincere gratitude to the reviewer for the valuable insights.
>
> ## Weakness:
>
> 1. best-of-N: The Best-of-N sampling strategy can be conceptualized as an output refinement algorithm that selects the highest-reward sample from multiple stochastic generations. Since our method does not modify the LLM's parameters, refining only the input prompt offers limited influence over the output distribution. Consequently, the outputs generated by our approach are likely contained within the set of candidates producible through repeated Best-of-N sampling, implying that Best-of-N represents a performance upper bound for our method. However, a key advantage of our approach lies in its inference efficiency: it achieves competitive results with just a single forward pass, significantly reducing computational cost compared to the multiple sampling steps required by Best-of-N.
>
> 2. BPO: While BPO is also a lightweight method that refines the input via a small model to enhance final output quality, its refinement and generation stages remain inherently disjointed. We posit that this decoupling can lead to refined prompts that are suboptimally aligned with the target LLM, thereby explaining its inferior performance compared to our integrated approach.
>
> DPO: A distributional shift exists between the training and test datasets. Both our method and BPO solely refine the input while keeping the generative LLM entirely unchanged. Consequently, although the data distributions differ, the preserved capabilities of the LLM provide a guaranteed level of performance on the test set. In contrast, since DPO is fine-tuned on a dataset that differs significantly from the test distribution, its performance is not guaranteed to surpass that of prompt-based methods.
>
> ## Question:
>
> Large-scale models, including Llama-8B, Mistral-7B, and Qwen-14B, are employed as black-box generators. In contrast, smaller models such as Llama-1B and Llama-3B serve as refinement models. This design intentionally leverages efficient, lightweight refinement models to enhance the performance of the high-capacity, fixed black-box generators through optimized input conditioning.
>
> We appreciate your additional suggestions regarding presentation and expression. We will carefully incorporate these recommendations into our revisions.

---

### Official Review · Reviewer_TdUh · 2025-11-03

**Soundness:** 3
**Presentation:** 2
**Contribution:** 2
**Rating:** 2
**Confidence:** 2

**Summary:**

This paper proposes a method for LLM alignment that bypasses traditional parameter-updating approaches like RLHF and DPO. The core idea is to use a separate "Refiner" model that transforms user queries into refined inputs that better align with desired outputs from black-box generation models. The method employs CMA-ES (Covariance Matrix Adaptation Evolution Strategy) to optimize the refiner's behavior based on feedback from the generation model, combined with posterior regularization using preference data.

**Strengths:**

* Practical problem setup that addresses the difficulty of finetuning closed-source models
* Novel viewpoint of alignment by modifying inputs to achieve the alignment goal. The latent variable approach is also generally interesting.

**Weaknesses:**

* Limited interpretation of latent variables. The paper claims it represents "user preference with diversity" and "reasoning paths," but provides no theoretical or empirical evidence to support this.
* Presentation and Clarity Issues. Figure 2 is confusing to understand given the fact that math notions appears much later.
* Using only 256 samples for CMA-ES optimization seems insufficient for robust optimization, especially given CMA-ES's sample complexity. It would be good to have variance (or similar stuff) reported.
* Technical confusion. Please see questions.

**Questions:**

* The constraint function B(.) seems crucial but is barely explained. How was this specific form derived?
* Given the uncertainty of latent variable z, why is the latent variable z necessary? Can we get rid of z?

---

> ### Author Response · Authors · 2025-11-20
> **Discussions on the reviews**
>
> We extend our sincere gratitude to the reviewer for the valuable insights.
>
> ## Latent variable z
>
> In our algorithmic framework, the latent variable *z* serves an essential function. Specifically, *z*—along with its positional markers [Zs] and [Ze]—acts as the anchor point for the CMA-ES optimization process. By fine-tuning this latent representation, we effectively transmit signals derived from CMA-ES back to the refinement model.
>
> A direct optimization path from input *x* to reconstructed x' would be highly challenging, as the feedback from the output log-probabilities is difficult to propagate back through discrete text space. Here, *z* acts as a continuous stylistic embedding that encapsulates diverse preferences. The CMA-ES feedback of different preferences dynamically adjusts *z*, which then guides the decoding process to produce a refined input aligned with the target objectives.
>
> ## Constraint function B(.)
>
> The design of the B(·) module is intended to preserve the readability of the refined text generated by the refinement model. Although the refinement model is based on a language model, parameter optimization via CMA-ES—unlike conventional gradient-based methods such as SFT—does not inherently guarantee output coherence or linguistic quality. To address this, we incorporate a gradient projection strategy into the CMA-ES optimization process. This strategy projects the update direction suggested by CMA-ES onto the gradient direction derived from a standard gradient-based method. As a result, optimization stability and output readability are maintained. In Equation (13), the term $\nabla E$ denotes the gradient obtained from the gradient-based method, while the preceding element represents the inner product that quantifies the alignment between the CMA-ES feedback signal and this gradient direction.
>
> We appreciate your additional suggestions regarding presentation and expression. We will carefully incorporate these recommendations into our revisions.

---

### Author Response · Authors · 2025-12-02

We thank the chair and the reviewers for their insightful comments. The primary concerns in the reviews appear to revolve around the justification of our methodological approach. We address these points in detail.

***Regarding Innovation***: Our method is motivated by a fundamental mismatch in traditional pipelines that treat input refinement and generation as independent stages. Conventional approaches, such as BPO, train a refinement model on refined data pairs in isolation and subsequently apply it to any LLM. This process does not optimize the refinement model with respect to the specific generation process of a target LLM. In practice, however, different LLMs exhibit varying sensitivities and preferences to prompts; a prompt effective for one model may not transfer optimally to another. Therefore, our work, which jointly optimizes refinement for a specific generator, addresses a genuine need and presents a novel contribution.

***Regarding Our Solution***: To achieve joint optimization of input refinement and generation, we propose a black-box optimization approach based on CMA-ES. This strategy allows us to keep the generation LLM entirely frozen, making it efficient and particularly suitable for API-based or proprietary LLMs where parameter updates are infeasible. To effectively apply CMA-ES, an anchor within the refinement model is required. We naturally employ a latent variable z as this anchor, enabling direct optimization. Since the refinement model itself is a sequence-to-sequence model, directly perturbing z can harm output readability. To mitigate this, we introduce the function B(·), which guides the optimization of z along a gradient-aligned direction—a design inspired by gradient-based training, which inherently preserves linguistic coherence.

***Regarding Experimental Results***: Reviewers raised questions about our performance compared to BPO and DPO. We clarify that BPO, as implemented in our comparison, does not involve full fine-tuning of the LLM (there may be confusion with other similarly named methods by the reviewer). DPO, while powerful, typically excels when test data follows the same distribution as its training data. In our experimental setup, all methods are evaluated on out-of-distribution test data. Critically, DPO modifies the base LLM's parameters, whereas our method and BPO leave the generator unchanged. This inherent advantage in generalization to unseen distributions explains why our approach legitimately outperforms DPO in this setting We have also updated the paper and provided a detailed explanation of the results for our method, BPO, and DPO at line 458.

***Figure 2***:
Regarding Figure 2, which presents the main framework of our proposed algorithm, we have included a detailed description of each component in the corresponding caption.

We respectfully request that the chair consider our detailed response for a more objective assessment of our work.

---

> ### Author Response · Authors · 2025-12-04
>
> # Point-by-point restatements to each reviewer
>
> ## 1. Reviewer ***TdUh***:
> a. ***Latent variable***: In our design, we employ the CMA-ES algorithm to jointly optimize input refinement and generation, with the latent variable — including its positional marker tokens — serving as an anchor for CMA-ES. Optimization is achieved through fine-tuning the representations associated with z.
>
> b. ***Figure 2***: We have included a detailed description of each component in the corresponding caption.
>
> c. ***256 samples***: In designing the CMA-ES optimization, we do not adopt the original search strategy. Instead, a constraint function B(·) is introduced to map the optimization direction along the gradient direction, thereby reducing the optimization difficulty. Considering training efficiency, a sample size of 256 is selected for implementation.
>
> d. ***Constraint function B(.)***: The design of the B(·) module is intended to preserve the readability of the refined text generated by the refinement model. This strategy projects the update direction suggested by CMA-ES onto the gradient direction derived from a standard gradient-based method.
>
> ## 2. Reviewer ***jAdW***:
> a. ***Figure 2***: We have included a detailed description of each component in the corresponding caption
>
> b. ***best-of-N***: Our method does not modify the LLM's parameters, refining only the input prompt offers limited influence over the output distribution. Consequently, the outputs generated by our approach are likely contained within the set of candidates producible through repeated Best-of-N sampling, implying that Best-of-N represents a performance upper bound for our method.
>
> c. ***BPO***: BPO is also a lightweight method that refines the input via a small model to enhance final output quality, its refinement and generation stages remain inherently disjointed.
>
> d. ***DPO***: DPO is fine-tuned on a dataset that differs significantly from the test distribution, its performance is not guaranteed to surpass that of prompt-based methods, i,e., our method and BPO.
>
> e. ***Choose the model***: Large-scale models, including Llama-8B, Mistral-7B, and Qwen-14B, are employed as black-box generators. In contrast, smaller models such as Llama-1B and Llama-3B serve as refinement models. This design intentionally leverages efficient, lightweight refinement models to enhance the performance of the high-capacity, fixed black-box generators through optimized input conditioning.
>
> ## 3. Reviewer ***xkVV***:
> a. ***Prior works***: Conventionally, most input refinement methods decouple the refinement process from LLM generation, treating them as independent steps. However, our algorithm bridges these two processes by leveraging the CMA-ES optimizer.
>
> b. ***Revise both input and output***: Although our method does not explicitly refine the output, it effectively propagates optimization signals from the output back to the input refinement via CMA-ES, leading to a substantively enhanced refinement process.
>
> c. ***Posterior regularization***: The overarching objective remains to leverage the X-Y alignment present in the data to develop a robust model with strong generalization capabilities. Therefore, achieving this X-Y alignment serves as a foundational premise in our algorithmic design.
>
> d. ***More detailed explanation***: The input x is first fed into the language model to obtain its hidden state, denoted as z. This hidden state z, which shares the same dimensionality as the model's embedding space, is then treated as a continuous embedding and reprojected into the language model. The model subsequently decodes this representation to reconstruct x'. In this process, the special tokens [Zs] and [Ze] are used to demarcate the positional boundaries of z within the sequence.
>
> ## 4. Reviewer ***wsCV***:
> a. ***Novelty***: Our principal contribution lies in establishing a cohesive link between the traditionally separate processes of refinement and generation.
>
> b. ***BPO***: BPO is also a lightweight method that refines the input via a small model to enhance final output quality, its refinement and generation stages remain inherently disjointed.
>
> c. ***DPO***: DPO is fine-tuned on a dataset that differs significantly from the test distribution, its performance is not guaranteed to surpass that of prompt-based methods, i,e., our method and BPO.

---

### Meta-Review · Area_Chair_8vuf · 2026-01-04

**Summary:**

All reviewers hold the concerns about the novelties, contributions, and experiment results, and give the negative scores.

**Reviewer Concerns:**

All reviewers hold the concerns about:

1. Weak novelty - the approach is too similar to existing prompt refinement methods
2. Poor presentation - figures, tables, and explanations need significant improvement
3. Questionable experimental results - counterintuitive findings require better explanation and analysis
4. Limited contribution - joint optimization of refinement and generation, while interesting, isn't sufficiently justified or impactful

**Reviewer Scores:**

I do not think any reviewer will change the score to positive, because they hold critical concerns about this paper.

---

### Decision · Program_Chairs · 2026-01-26

Reject